# RELIABLE UNCERTAINTY ESTIMATES IN NEURAL NETWORKS USING NOISE CONTRASTIVE PRIORS

## ABSTRACT

Obtaining reliable uncertainty estimates of neural network predictions is a long standing challenge. Bayesian neural networks have been proposed as a solution, but it remains open how to specify their prior. In particular, the common practice of a standard normal prior in weight space imposes only weak regularities, causing the function posterior to possibly generalize in unforeseen ways on inputs outside of the training distribution. We propose noise contrastive priors (NCPs) to obtain reliable uncertainty estimates. The key idea is to train the model to output high uncertainty for data points outside of the training distribution. NCPs do so using an input prior, which adds noise to the inputs of the current mini batch, and an output prior, which is a wide distribution given these inputs. NCPs are compatible with any model that can output uncertainty estimates, are easy to scale, and yield reliable uncertainty estimates throughout training. Empirically, we show that NCPs prevent overfitting outside of the training distribution and result in uncertainty estimates that are useful for active learning. We demonstrate the scalability of our method on the flight delays data set, where we significantly improve upon previously published results.

## 1 INTRODUCTION

Many successful applications of neural networks (Krizhevsky et al., 2012; Sutskever et al., 2014; van den Oord et al., 2016) are in restricted settings where predictions are only made for inputs similar to the training distribution. In real-world scenarios, neural networks can face truly novel data points during inference, and in these settings it can be valuable to have good estimates of the model's uncertainty. For example, in healthcare, reliable uncertainty estimates can prevent overconfident decisions for rare or novel patient conditions (Schulam and Saria, 2015). Similarly, autonomous agents that actively explore their environment can use uncertainty estimates to decide what data points will be most informative.

Epistemic uncertainty describes the amount of missing knowledge about the data generating function. Uncertainty can in principle be completely reduced by observing more data points at the right locations and training on them. In contrast, the data generating function may also have inherent randomness, which we call aleatoric noise. This noise can be captured by models outputting a distribution rather than a point prediction. Obtaining more data points allows the noise estimate to move closer to the true value, which is usually different from zero. For active learning, it is crucial to separate the two types of randomness: we want to acquire labels in regions of high uncertainty but low noise (MacKay, 1992a).

Bayesian analysis provides a principled approach to modeling uncertainty in neural networks (Denker et al., 1987; MacKay, 1992b). Namely, one places a prior over the network's weights and biases. This effectively places a distribution over the functions that the network represents, capturing uncertainty about which function best fits the data. Specifying this prior remains an open challenge. Common practice is to use a standard normal prior in weight space, which imposes weak shrinkage regularities analogous to weight decay. It is neither informative about the induced function class nor the data (e.g., it is sensitive to parameterization).

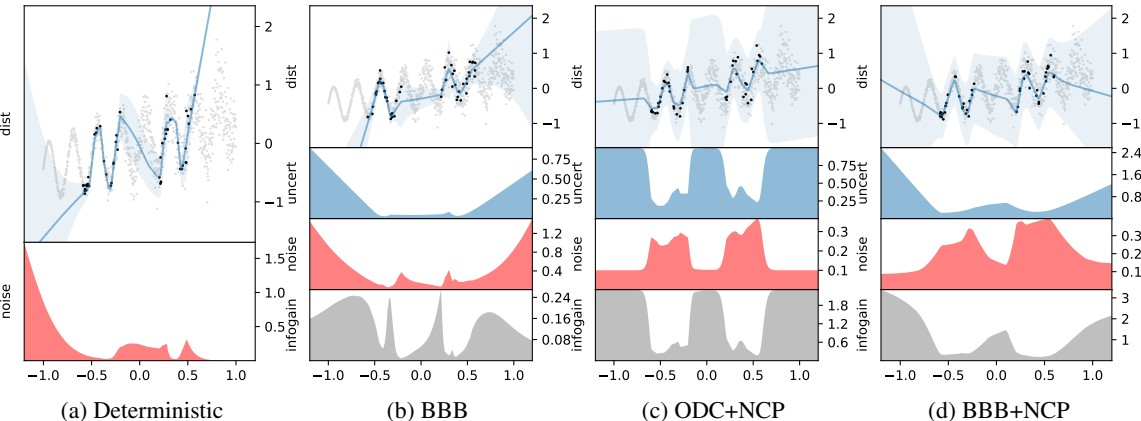

Figure 1: Predictive distributions on a low-dimensional active learning task. The predictive distributions are visualized as mean and two standard deviations shaded. They decompose into epistemic uncertainty ▉ and aleatoric noise ▉. Data points are only available within two bands, and are selected using the expected information gain ▉. **(a)** A deterministic network conflates uncertainty as part of the noise and is overconfident outside of the data distribution. **(b)** A variational Bayesian neural network with standard normal prior represents uncertainty and noise separately but is overconfident outside of the training distribution. **(c)** On the OOD classifier model, NCP prevents overconfidence. **(d)** On the Bayesian neural network, NCP produces smooth uncertainty estimates that generalize well to unseen data points. Models trained with NCP also separate uncertainty and noise well. The experimental setup is described in Section 5.1.

This can cause the induced function posterior to generalize in unforeseen ways on out-of-distribution (OOD) inputs, which are inputs outside of the distribution that generated the training data.

Motivated by these challenges, we introduce noise contrastive priors (NCPs), which encourage uncertainty outside of the training distribution through a loss in data space. NCPs are compatible with any model that represents functional uncertainty as a random variable, are easy to scale, and yield reliable uncertainty estimates that show significantly improved active learning performance.

## 2 NOISE CONTRASTIVE PRIORS

Specifying priors is intuitive for small probabilistic models, where each variable typically has a clear interpretation (Blei, 2014). It is less intuitive for neural networks, where the parameters serve more as adaptive basis coefficients in a nonparametric function. For example, neural network models are non-identifiable due to weight symmetries that yield the same function (Müller and Insua, 1998). This makes it difficult to express informative priors on the weights, such as expressing high uncertainty on unfamiliar examples.

**Data priors**  Unlike a prior in weight space, a *data prior* lets one easily express informative assumptions about input-output relationships. Here, we use the example of a prior over a labeled data set $\{x, y\}$, although the prior can also be on $x$ and another variable in the model that represents uncertainty and has a clear interpretation. The prior takes the form $p_{\text{prior}}(x, y) = p_{\text{prior}}(x) \, p_{\text{prior}}(y \mid x)$, where $p_{\text{prior}}(x)$ denotes the *input prior* and $p_{\text{prior}}(y \mid x)$ denotes the *output prior*.

To prevent overconfident predictions, a good input prior $p_{\text{prior}}(x)$ should include OOD examples so that it acts beyond the training distribution. A good output prior $p_{\text{prior}}(y \mid x)$ should be a high-entropy distribution, representing high uncertainty about the model output given OOD inputs.

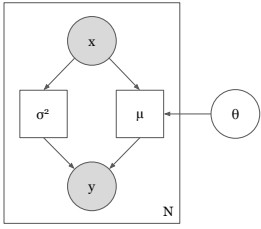
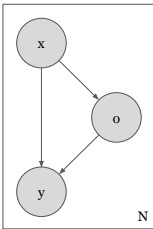

(a) Bayesian neural network       (b) Out-of-distribution classifier

Figure 2: Graphical representations of the two uncertainty-aware models we consider. Circles denote random variables, squares denote deterministic variables, shading denotes observations during training. **(a)** The Bayesian neural network captures a belief over parameters for the predictive mean, while the predictive variance is a deterministic function of the input. In practice, we only use weight uncertainty for the mean's output layer and share earlier layers between the mean and variance. **(b)** The out-of-distribution classifier model uses a binary auxiliary variable $o$ to determine if a given input is out-of-distribution; given its value, the output is drawn from either a neural network prediction or a wide output prior.

**Generating OOD inputs**  Exactly generating OOD data is difficult. A priori, we must uniformly represent the input domain. A posteriori, we must represent the complement of the training distribution. Both distributions are typically uniform over infinite support, making them ill-defined. To estimate OOD inputs, we develop an algorithm inspired by noise contrastive estimation (Gutmann and Hyvärinen, 2010a; Mnih and Kavukcuoglu, 2013), where a complement distribution is approximated using random noise.

A hypothesis of our work is that in practice it is enough to encourage high uncertainty output near the *boundary* of the training distribution, and that this effect will propagate to the entire OOD space. This hypothesis is backed up by previous work (Lee et al., 2017) as well as our experiments (see Figure 1). This means we no longer need to sample arbitrary OOD inputs. It is enough to sample OOD points that lie close to the boundary of the training distribution, and to apply our desired prior at those points.

**Loss function**  Noise contrastive priors are data priors that are enforced on both training inputs $x$ and inputs $\tilde{x}$ perturbed by noise. For example, in binary and categorical input domains, we approximate OOD inputs by randomly flipping the features to different classes with a certain probability. For continuous valued inputs $x$, we can use additive Gaussian noise to obtain noised up inputs $\tilde{x} = x + \epsilon$. This expresses the noise contrastive prior where inputs are distributed according to the convolved distribution,

$$p_{\text{prior}}(\tilde{x}) = \int_x p_{\text{train}}(x)\,\text{Normal}(\tilde{x} - x \mid \mu_x, \sigma_x^2)\,dx \qquad p_{\text{prior}}(\tilde{y} \mid \tilde{x}) = \text{Normal}(\mu_y, \sigma_y^2). \qquad (1)$$

The variances $\sigma_x^2$ and $\sigma_y^2$ are hyperparameters that tune how far from the boundary we sample, and how large we want the output uncertainty to be. We choose $\mu_x = 0$ to apply the prior equally in all directions from the data manifold. The output mean $\mu_y$ determines the default prediction of the model outside of the training distribution, for example $\mu_y = 0$. We set $\mu_y = y$ which corresponds to data augmentation (Matsuoka, 1992; An, 1996), where a model is trained to recover the true labels from perturbed inputs. This way, NCP makes the model uncertain while still trying to generalize to OOD inputs.

For training, we minimize the loss function

$$\begin{aligned}\mathcal{L}(\theta) = \ &\text{E}_{p_{\text{train}}(x)}\big[D_{\text{KL}}[p_{\text{train}}(y \mid x) \,\|\, p_{\text{model}}(y \mid x, \theta)]\big] \\ &+ \gamma \text{E}_{p_{\text{prior}}(\tilde{x})}\big[D_{\text{KL}}[p_{\text{prior}}(\tilde{y} \mid \tilde{x}) \,\|\, p_{\text{model}}(\tilde{y} \mid \tilde{x}, \theta)]\big].\end{aligned} \qquad (2)$$

The first term represents typical maximum likelihood, in which one minimizes the KL divergence to the empirical training distribution $p_{\text{train}}(y \mid x)$ over training inputs. The second term is added by our method: it represents the analogous term on a data prior. The hyperparameter $\gamma$ sets the relative trade-off between them.

**Interpretation as function prior**    The noise contrastive prior can be interpreted as inducing a function prior. This is formalized through the prior predictive distribution,

$$p(y \mid x) = \int p_{\text{model}}(y \mid x, \theta) \, p_{\text{model}}(\theta \mid \tilde{x}, \tilde{y}) \, p_{\text{prior}}(\tilde{x}, \tilde{y}) \, d\theta \, d\tilde{x} \, d\tilde{y}. \tag{3}$$

The distribution marginalizes over network parameters $\theta$ as well as data fantasized from the data prior. The distribution $p(\theta \mid \tilde{x}, \tilde{y})$ represents the distribution of model parameters after fitting the prior data. That is, the belief over weights is shaped to make $p(y \mid x)$ highly variable. This parameter belief causes uncertain predictions outside of the training distribution, which we could not specify in weight space directly.

Because network weights are constrained to fit the data prior, the prior acts as "pseudo-data." This is similar to classical work on conjugate priors: a $\text{Beta}(\alpha, \beta)$ prior on the probability of a Bernoulli likelihood implies a Beta posterior, and if the posterior mode is chosen as an optimal parameter setting, then the prior translates to $\alpha - 1$ successes and $\beta - 1$ failures. It is also similar to pseudo-data in sparse Gaussian processes (Quiñonero-Candela and Rasmussen, 2005).

Data priors encourage learning parameters that not only capture the training data well but also the prior data. In practice, we can combine NCP with other priors, for example the typical standard normal prior in weight space for Bayesian neural networks, although we did not find this necessary in our experiments.

## 3    BAYESIAN NEURAL NETWORKS WITH NCP

Noise contrastive priors are applicable to any model that represents uncertainty in a random variable. The NCP can then be added to that random variable to make the model uncertain on OOD inputs. In this section, we apply NCP to a Bayesian neural network (BNN) trained via variational inference. Blundell et al. (2015) introduce such a model under the name Bayes by Backprop (BBB) that uses a standard normal prior in weight space. We extend this model with a NCP on the mean predicted by the neural network.

Consider a regression task with data $\{x, y\}$ that we model as $p(y \mid x, \theta) = \text{Normal}(\mu(x), \sigma^2(x))$ with mean and variance predicted by a neural network from the inputs. This model is heteroskedastic, meaning that it can predict a different aleatoric noise amount for every point in the input space. We use a weight prior for only the output layer (Lázaro-Gredilla and Figueiras-Vidal, 2010; Calandra et al., 2014) that predicts the mean, resulting in the model

$$\theta \sim \text{Normal}(0, 0.1) \qquad y \sim \text{Normal}(\mu(x, \theta), \sigma^2(x)). \tag{4}$$

We do not model uncertainty about the noise estimate, as this is not required for the approximation for the Gaussian expected information gain (MacKay, 1992a) that we use to acquire labels. Therefore, the distribution of the mean induced by the weight prior, $q(\mu(x)) = \int \mu(x, \theta) q_\phi(\theta) \, d\theta$, represents the model's epistemic uncertainty. Note that this is different from the predictive distribution, which combines both uncertainty and noise. We place an NCP on the distribution of the mean, resulting in the loss function

$$\mathcal{L}(\phi) = -\mathbb{E}_{q_\phi(\theta)}[\ln p(y \mid x, \theta)] + \beta D_{\text{KL}}[q_\phi(\theta) \parallel p(\theta)] + \underbrace{\gamma D_{\text{KL}}[\text{Normal}(\mu_\mu, \sigma_\mu^2) \parallel q(\mu(\tilde{x}))]}_{\text{NCP loss}}. \tag{5}$$

Here, $\tilde{x}$ are the perturbed inputs and $q_\phi(\theta)$ forms an approximate posterior over weights.[1] Because we only use the weight belief for the linear output layer, we can compute the KL-divergence of the NCP loss analytically. In other models, it could be estimated using samples.

The loss function applies weight regularization in order for network weights to regress to a standard normal prior; like other regularization techniques, this assists in improving the network's generalization in-distribution.

---

[1] To derive the loss, set $p(y \mid x, \theta) = \mathbb{E}_{q_\phi(\theta)}[p(y \mid x, \theta)]$ in Equation 2 and apply Jensen's inequality (Blundell et al., 2015; Higgins et al., 2016).

The NCP loss encourages the network's generalization OOD by matching the mean distribution to the output prior. Minimizing the KL divergence to a wide output prior results in high uncertainty on OOD inputs, so the model will explore these data points during active learning.

In practice, we find that NCP is sufficient as a prior for the BNN and set $\beta = 0$. The appendix (Appendix B includes an alternative interpretation explaining why NCP might be sufficient, which represents the weight space KL-divergence in data space after a change of variables.

## 4  RELATED WORK

**Priors for neural networks**  Classic work has investigated entropic priors (Buntine and Weigend, 1991) and hierarchical priors (MacKay, 1992b; Neal, 2012; Lampinen and Vehtari, 2001). More recently, Depeweg et al. (2018) introduce networks with latent variables in order to disentangle forms of uncertainty, and Flam-Shepherd et al. (2017) propose general-purpose weight priors based on approximating Gaussian processes. Other works have analyzed priors for compression and model selection (Ghosh and Doshi-Velez, 2017; Louizos et al., 2017). Instead of a prior in weight space (or latent inputs as in Depeweg et al. (2018)), NCPs take the functional view by imposing explicit regularities in terms of the network's inputs and outputs. Malinin and Gales (2018) propose prior networks to avoid an explicit belief over parameters for classification tasks.

**Input and output regularization**  There is classic work on adding noise to inputs for improved generalization (Matsuoka, 1992; An, 1996; Bishop, 1995). For example, denoising autoencoders (Vincent et al., 2008) encourage reconstructions given noisy encodings. Output regularization is also a classic idea from the maximum entropy principle (Jaynes, 1957), where it has motivated label smoothing (Szegedy et al., 2016) and entropy penalties (Pereyra et al., 2017). Also related is virtual adversarial training (Miyato et al., 2015), which includes examples that are close to the current input but cause a maximal change in the model output, and mixup (Zhang et al., 2018), which includes examples under the vicinity of training data. These methods are orthogonal to NCPs: they aim to improve generalization from finite data within the training distribution (interpolation), while we aim to improve uncertainty estimates outside of the training distribution (extrapolation).

**Classifying out-of-distribution inputs**  A simple approach for neural network uncertainty is to classify whether data points belong to the data distribution, or are OOD (Hendrycks and Gimpel, 2017). This is core to noise contrastive estimation (Gutmann and Hyvärinen, 2010b), a training method for intractable probabilistic models. More recently, Lee et al. (2017) introduce a GAN to generate OOD samples, and Liang et al. (2018) add perturbations to the input, applying an "OOD detector" to improve softmax scores on OOD samples by scaling the temperature. Extending these directions of research, we connect to Bayesian principles and focus on uncertainty estimates that are useful for active data acquisition.

## 5  EXPERIMENTS

To demonstrate their usefulness, we evaluate NCPs on various tasks where uncertainty estimates are desired. Our focus is on active learning for regression tasks, where only few targets are visible in the beginning, and additional targets are selected regularly based on an acquisition function. We use two data sets: a toy example and a large flights data set. We also evaluate how sensitive our method is to the choice of input noise. Finally, we show that NCP scales to large data sets by training on the full flights data set in a passive learning setting. Our implementation uses TensorFlow Probability (Dillon et al., 2017; Tran et al., 2016) and is open-sourced at `https://<hidden-for-review>`.

We compare four neural network models, all using leaky ReLU activations (Maas et al., 2013) and trained using Adam (Kingma and Ba, 2014). The four models are:

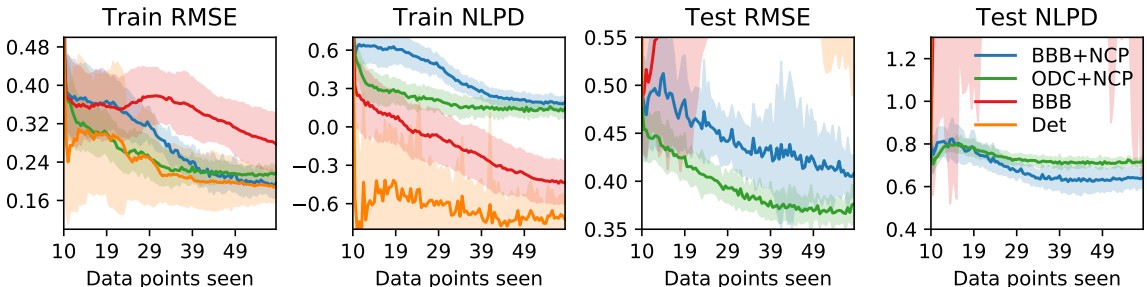

Figure 3: Active learning on the 1-dimensional regression problem, mean and standard deviation over 20 seeds. The test root mean squared error (RMSE) and negative log predictive density (NLPD) of the models trained with NCP decreases during the active learning run, while the baseline models select less informative data and overfit. The deterministic network is barely visible in the plots as it overfits quickly. Figure 1 shows the predictive distributions of the models.

- **Deterministic neural network (Det)**  A neural network that predicts the mean and variance of a normal distribution. The name stands for *deterministic*, as there is no weight uncertainty.

- **Bayes by Backprop (BBB)**  A Bayesian neural network trained via gradient-based variational inference with a standard normal prior in weight space (Blundell et al., 2015; Kucukelbir et al., 2017). We use the same model as in Section 3 but without the NCP loss term.

- **Bayes by Backprop with noise contrastive prior (BBB+NCP)**  Bayes by Backprop with NCP on the predicted mean distribution as described in Section 3.

- **Out-of-distribution classifier with noise contrastive prior (OCD+NCP)**  An uncertainty classifier model described in Appendix A. It is a deterministic neural network combined with NCP which we use as a baseline alternative to Bayes by Backprop with NCP.

For active learning, we select new data points $\{x, y\}$ for which $x$ maximizes the expected information gain $\mathrm{E}_{q(y|x)}[D_{\mathrm{KL}}[q(\theta \mid x, y) \parallel q(\theta)]]$ under the model $q(y \mid x) = \int p(y \mid x, \theta) q(\theta)\, d\theta$. Intuitively, this objective function is higher where the model has high epistemic uncertainty and predicts low aleatoric noise.

We use an approximation from MacKay (1992a) for Gaussian posterior predictive distributions. Moreover, we place a softmax distribution on the information gain for all available data points and acquire labels by sampling with a temperature of $\tau = 0.5$ to get diversity when selecting batches of labels,

$$\{x_{\mathrm{new}}\} \sim p_{\mathrm{new}}(x) = \frac{1}{Z} \exp\left(\frac{1}{2\tau} \ln\left(1 + \mathrm{Var}[q(\mu(x))]/\sigma^2(x)\right)\right) = \frac{1}{Z}\left(1 + \mathrm{Var}[q(\mu(x))]/\sigma^2(x)\right), \quad (6)$$

where $\sigma^2(x)$ is the estimated aleatoric noise and $q(\mu(x))$ is the epistemic uncertainty projected into output space. Since our Bayesian neural networks only use a weight belief for the output layer, $\mathrm{Var}[q(\mu(x))]$ is Gaussian and can be computed in closed form. In general, it the epistemic part of the predictive variance would be estimated by sampling. In the classifier model, we use the OOD probability $p(o = 1|x)$ for this. For the deterministic neural network, we use $\mathrm{Var}[p(y \mid x)]$ as proxy since it does not output an estimate of epistemic uncertainty.

### 5.1 Low-dimensional active learning

For visualization purposes, we start with experiments on a 1-dimensional regression task that consists of a sine function with a small slope and increasing variance for higher inputs. Training data can be acquired within two bands, and the model is evaluated on all data points that are not visible to the model. This structured split

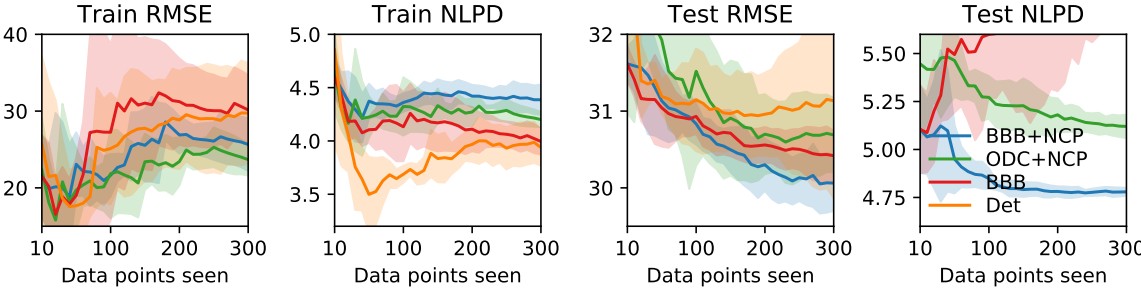

Figure 4: Active learning on the flights data set. The models trained with NCP achieve significantly lower negative log predictive density (NLPD) on the test set, and Bayes by Backprop with NCP achieves the lowest root mean squared error (RMSE). The test NLPD for the baseline models diverges as they overfit to the visible data points. Plots show mean and std over 10 runs.

between training and testing data causes a distributional shift at test time, requiring successful models to have reliable uncertainty estimates to avoid mispredictions for OOD inputs.

For this experiment, we use two layers of 200 hidden units, a batch size of 10, and a learning rate of $3 \times 10^{-4}$ for all models. NCP models use noise $\epsilon \sim \text{Normal}(0, 0.5)$. We start with 10 randomly selected initial targets, and select 1 additional target every 1000 epochs. Figure 3 shows the root mean squared error (RMSE) and negative log predictive density (NLPD) throughout learning. The two baseline models severely overfit to the training distribution early on when only few data points are visible. Models with NCP outperform BBB, which in turn outperforms Det. Figure 1 visualizes the models' predictive distributions at the end of training, showing that NCP prevents overconfident generalization.

## 5.2 ACTIVE LEARNING ON FLIGHT DELAYS

We consider the flight delay data set (Hensman et al., 2013; Deisenroth and Ng, 2015; Lakshminarayanan et al., 2016), a large scale regression benchmark with several published results. The data set has 8 input variables describing a flight, and the target is the delay of the flight in minutes. There are 700K training examples and 100K test examples. The test set has a subtle distributional shift, since the 100K data points temporally follow after the training data.

We use two layers with 50 units each, a batch size of 10, and a learning rate of $10^{-4}$. For NCP models, $\epsilon \sim \text{Normal}(0, 0.1)$. Starting from 10 labels, the models select a batch of 10 additional labels every 50 epochs. The 700K data points of the training data set are available for acquisition, and we evaluate performance on the typical test split. Figure 4 shows the performance for the visible data points and the test set respectively. We note that BBB and BBB+NCP show similar NLPD on the visible data points, but the NCP models generalize better to unseen data. Moreover, the Bayesian neural network with NCP achieves lower RMSE than the one without and the classifier based model achieves lower RMSE than the deterministic neural network. All uncertainty-based models outperform the deterministic neural network.

## 5.3 ROBUSTNESS TO NOISE PATTERNS

The choice of input noise might seem like a critical hyper parameter for NCP. In this experiment, we find that our method is robust to the choice of input noise. The experimental setup is the same as for the active learning experiment described in Section 5.2, but with uniform or normal input noise with different variance ($\sigma_x^2 \in \{0.1, 0.2, \cdots, 1.0\}$). For uniform input noise, this means noise is drawn from the interval $[-2\sigma_x, 2\sigma_x]$.

We observe that BBB+NCP is robust to the size of the input noise. NCP consistently improves RMSE for the tested noise sizes and yields the best NLPD for all noise sizes below 0.6. For our ODC baseline, we

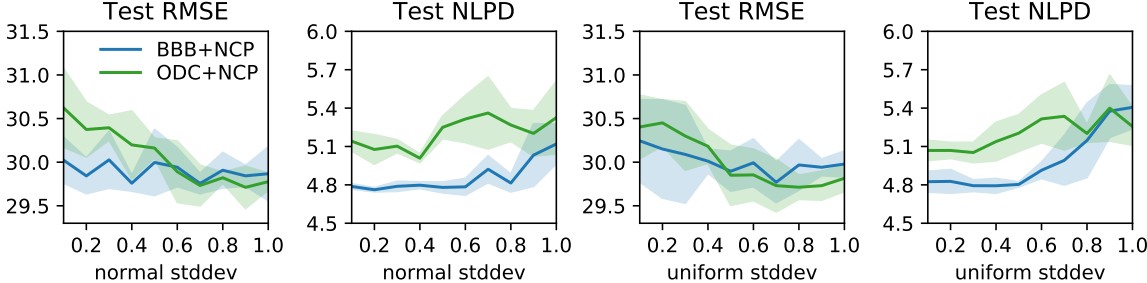

Figure 5: Robustness to different noise patterns. Plots show the final test performance on the flights active learning task (mean and stddev over 5 seeds). Lower is better. NCP is robust to the choice of input noise and improves over the baselines in all settings (compare Figure 4).

observe an intuitive trade-off: smaller input noise increases the regularization strength, leading to better NLPD but reduced RMSE. Robustness to the choice of input noise is further supported by the analogous experiment on toy data set, where above a small threshold (BBB+NCP $\sigma_x^2 \geq 0.3$ and ODC+NCP $\sigma_x^2 \geq 0.1$), NCP consistently performs well (Figure 6).

### 5.4   LARGE SCALE REGRESSION OF FLIGHT DELAYS

In addition to the active learning experiments, we perform a passive learning run on all 700K data points of the flights data set to explore the scalability of NCP. We use networks of 3 layers with 1000 units and a learning rate of $10^{-4}$. Table 1 compares the performance of our models to previously published results. We significantly improve state of the art performance on this data set.

## 6   DISCUSSION

We develop *noise contrastive priors* (NCPs), a prior for neural networks in data space. NCPs encourage network weights that not only explain the training data but also capture high uncertainty on OOD inputs. We show that NCPs offer strong improvements over baselines and scale to large regression tasks.

We focused on active learning for regression tasks, where uncertainty is crucial for determining which data points to select next. In future work it would be interesting to apply NCPs to alternative settings where uncertainty is important, such as image classification and learning with sparse or missing data. In addition, NCPs are only one form of a data prior, designed to encourage uncertainty on OOD inputs. Priors in data space can easily capture other properties such as periodicity or spatial invariance, and they may provide a scalable alternative to Gaussian process priors.

Table 1: Performance on all 700K data points of the flights data set. While uncertainty estimates are not necessary when a large data set that is similar to the test data set is available, it shows that our method scales easily to large data sets.

| Model | NLPD | RMSE |
|---|---|---|
| gPoE (Deisenroth & Ng 2015) | 8.1 | — |
| SAVIGP (Bonilla et al. 2016) | 5.02 | — |
| SVI GP (Hensman et al. 2013) | — | 32.60 |
| HGP (Ng & Deisenroth 2014) | — | 27.45 |
| MF (Lakshminarayanan et al. 2016) | 4.89 | 26.57 |
| BBB | **4.38** | **24.59** |
| BBB+NCP | **4.38** | **24.71** |
| ODC+NCP | **4.38** | **24.68** |

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

# A OOD CLASSIFIER MODEL WITH NCP

We showed how to apply NCP to a Bayesian neural network model that captures function uncertainty in a belief over parameters. An alternative approach to capture uncertainty is to make explicit predictions about whether an input is OOD. There is no belief over weights in this model. Figure 2b shows such a mixture model via a binary variable $o$,

$$
o \sim \text{Bernoulli}(\pi(x, \theta))
$$
$$
y \sim \begin{cases} \text{Normal}(\mu(x, \theta), \sigma^2(x, \theta)) & \text{if } o = 0 \\ \text{Normal}(\mu_y, \sigma_y^2) & \text{if } o = 1, \end{cases} \tag{7}
$$

where $p(o = 1 \mid x)$ is the OOD probability of $x$. If $o = 0$ ("in distribution"), the model outputs the neural network prediction. Otherwise, if $o = 1$ ("out of distribution"), the model uses a fixed output prior. The neural network weights $\theta$ are estimated using a point estimate, so we do not maintain a belief distribution over them.

The classifier prediction $p(o \mid x, \theta)$ captures uncertainty in this model. We apply the NCP $p(o \mid \tilde{x}, \theta) = \delta(o = 1 | \tilde{x}, \theta)$ to this variable, which assumes noised-up inputs to be OOD. During training on the data set, $\{x, y\}$ and $o = 0$ are observed, as training data are in-distribution by definition. Following Equation 2, the loss function is

$$
\begin{aligned}
\mathcal{L}(\theta) &= D_{\text{KL}}[p_{\text{train}}(y \mid x) \parallel p_{\text{model}}(y \mid x, o = 0, \theta)] + D_{\text{KL}}[p_{\text{prior}}(\tilde{o} \mid \tilde{x}) \parallel p_{\text{model}}(\tilde{o} \mid \tilde{x}, \theta)] \\
&= -\ln p(y, o = 0 \mid x, \theta) - \ln p(y, o = 1 \mid \tilde{x}, \theta) \\
&= -\ln \text{Normal}(y \mid \mu(x, \theta), \sigma^2(x, \theta)) - \ln \text{Bernoulli}(0 \mid \pi(x, \theta)) \underbrace{- \ln \text{Bernoulli}(1 \mid \pi(\tilde{x}, \theta))}_{\text{NCP loss}}.
\end{aligned} \tag{8}
$$

Analogously to the Bayesian neural network model in Section 3, we can either set $\mu_y, \sigma_y^2$ manually or use the neural network prediction for potentially improved generalization. In our experiments, we implement the OOD classifier model using a single neural network with two output layers that parameterize the Gaussian distribution and the binary distribution.

# B BNN WITH NCP USING REVERSE KL

In Section 3, we derived the Bayes by Backprop model with NCP by adding a forward KL-divergence from the mean prior to the model mean to the loss. An alternative derivation uses the fact that the KL-divergence is invariant to parameterization to replace the reverse KL-divergence in weight space by a KL-divergence in output space,

$$
\begin{aligned}
\text{E}_{p(x,y)}\big[\ln p(y \mid x)\big] &= \text{E}_{p(x,y)}\left[\ln \int p(y \mid x, \theta) p(\theta) \frac{q(\theta)}{q(\theta)} \, d\theta\right] \\
&\geq \text{E}_{p(x,y)}\left[\int q(\theta) \ln p(y \mid x, \theta) \frac{p(\theta)}{q(\theta)} \, d\theta\right] \\
&= \text{E}_{p(x,y)}\big[\text{E}_{q(\theta)}[\ln p(y \mid x, \theta)] - D_{\text{KL}}[q(\theta) \parallel p(\theta)]\big] \\
&= \text{E}_{p(x,y)}\big[\text{E}_{q(\theta)}[\ln p(y \mid x, \theta)] - \text{E}_{p(\tilde{x}|x)}[D_{\text{KL}}[q(\theta) \parallel p(\theta)]]\big] \\
&\approx \text{E}_{p(x,y)}\big[\text{E}_{q(\theta)}[\ln p(y \mid x, \theta)] - \text{E}_{p(\tilde{x}|x)}[D_{\text{KL}}[q(\mu(\tilde{x})) \parallel p(\mu(\tilde{x}) \mid x)]]\big],
\end{aligned} \tag{9}
$$

where $p(\mu(\tilde{x})) = \int \mu(\tilde{x}, \theta) p(\theta) \, d\theta$ and $q(\mu(\tilde{x})) = \int \mu(\tilde{x}, \theta) q(\theta) \, d\theta$ are the distributions of the predicted mean induces by the weight beliefs. As a result, instead of specifying a prior in weight space, we can specify a prior in output space.

Above, we reparameteterized the KL in weight space as a KL in output space; by the change of variables, this is equivalent if the mapping $\mu(\cdot, \theta)$ is continuous and 1-1 with respect to $\theta$. This assumption does not hold for neural nets as multiple parameter vectors can lead to the same predictive distribution, thus the approximation above. A compact reparameterization of the neural network (equivalence class of parameters) would make this an equality.

## C ROBUSTNESS EXPERIMENT ON TOY DATASET

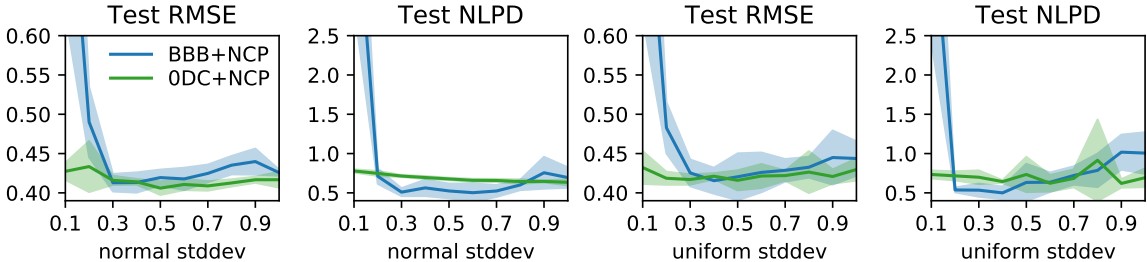

Figure 6: Robustness to different noise patterns. Plots show the final test performance on the low-dimensional active learning task (mean and stddev over 5 seeds). Lower is better. The baseline performances are RMSE: BBB ($0.75 \pm 0.31$), Det ($1.46 \pm 0.64$) and NLPD: BBB ($10.29 \pm 8.05$), Det ($1.3 \times 10^8 \pm 1.7 \times 10^8$). NCP works with both Gaussian and uniform input noise $\epsilon$ and is robust to $\sigma_x^2$.

## D RELATED ACTIVE LEARNING WORK

Active learning is often employed in domains where data is cheap but labeling is expensive, and is motivated by the idea that not all data points are equally valuable when it comes to learning (Settles, 2009; Dasgupta, 2004). Active learning techniques can be coarsely grouped into three categories. Ensemble methods (Seung et al., 1992; McCallumzy and Nigamy, 1998; Freund et al., 1997) generate queries that have the greatest disagreement between a set of classifiers. Error reduction approaches incorporate the select data based on the predicted reduction in classifier error based on information (MacKay, 1992a), Monte Carlo estimation (Roy and McCallum, 2001), or hard-negative example mining (Sung, 1994; Rowley et al., 1998).

Uncertainty-based techniques select samples for which the classifier is most uncertain. Approaches include maximum entropy (Joshi et al., 2009), distance from the decision boundary (Tong and Koller, 2001), pseudo labelling high confidence examples (Wang et al., 2017), and mixtures of information density and uncertainty measures (Li and Guo, 2013). Within this category, the area most related to our work are Bayesian methods. Kapoor et al. (2007) estimate expected improvement using a Gaussian process. Other approaches use classifier confidence (Lewis and Gale, 1994), predicted expected error (Roy and McCallum, 2001), or model disagreement (Houlsby et al., 2011). Recently, Gal et al. (2017) applied a convolutional neural network with dropout uncertainty to images.

