# OpenReview forum: "Reliable Uncertainty Estimates in Deep Neural Networks using Noise Contrastive Priors"
_ICLR.cc/2019/Conference_

### Official Review · AnonReviewer3 · 2018-11-01
**Interesting paper; being more careful about experiments would strengthen it further.**

**Rating:** 6
**Confidence:** 4

**Review:**

The paper considers the problem of obtaining reliable predictive uncertainty estimates. The authors propose noise contrastive priors — the idea being to explicitly encourage high uncertainties for out of distribution (OOD) data through a loss in the data space.  OOD data is simulated by adding noise to existing data and the model is trained to maximize the likelihood wr.t. training data while being close in the KL sense to a (wide) conditional prior p(y | x) on the OOD responses (y).  The authors demonstrate that the procedure leads to improved uncertainty estimates on toy data and can better drive active learning on a large flight delay dataset.

The paper is well written and makes for a nice read. I like the idea of using “pseudo” OOD data for encouraging better behaved uncertainties away from the data. It is nice to see that even simple schemes for generating OOD data (adding iid noise) lead to improved uncertainty estimates.

My main concern about this work stems from not knowing how sensitive the recovered uncertainties are to the OOD data generating mechanism and the parameters thereof. The paper provides little evidence to conclude one way or the other.  The detailed comments below further elaborate on this concern.

Detailed Comments:
a) I like the sensitivity analysis presented in Figure 4, and it does show for the 1D sine wave the method is reasonably robust to the choice of \sigma_x. However, it is unclear how problem dependent the choice of sigma_x is. From the experiments, it seems that \sigma_x needs to be carefully chosen for different problems, \sigma^2_x < 0.3 seems to not work very well for BBB + NCP for the 1D sine data, but for the flight delay data \sigma^2_x is set to 0.1 and seems to work well. How was \sigma_x chosen for the different experiments?

b) It is also interesting that noise with a shared scale is used for all 8 dimensions of the flight dataset. Is this choice mainly governed by convenience — easier to select one hyper-parameter rather than eight?

c) Presumably, the predictive uncertainties are also strongly affected by both the weighting parameter \gamma and the prior variance sigma^2_y . How sensitive are the uncertainties to these and how were these values chosen for the experiments presented in the paper?

d) It would be really interesting to see how well the approach extends to data with more interesting correlations. For example, for image data would using standard data-augmentation techniques (affine transformations) for generating OOD data help over adding iid noise. In general, it would be good to have at least some empirical validation of the proposed approach on moderate-to-high dimensional data (such as images).

==============
Overall this is an interesting paper that could be significantly strengthened by addressing the comments above and a more careful discussion of how the procedure for generating OOD data affects the corresponding uncertainties.

---

> ### Author Response · Authors · 2018-11-27
> **Accurate summary, hyper parameter choice, new sensitivity analysis**
>
> Thank you very much for your review.
>
> [The paper is well written and makes for a nice read. I like the idea of using “pseudo” OOD data for encouraging better behaved uncertainties away from the data. It is nice to see that even simple schemes for generating OOD data (adding iid noise) lead to improved uncertainty estimates.]
>
> Thank you. We were positively surprised by these results, too.
>
> [a) I like the sensitivity analysis presented in Figure 4, and it does show for the 1D sine wave the method is reasonably robust to the choice of \sigma_x. However, it is unclear how problem dependent the choice of sigma_x is. [...] How was \sigma_x chosen for the different experiments?]
>
> For our experiments, we manually tuned \sigma_x to work for both our BBB+NCP model and the ODC+NCP baseline model. We later performed the sensitivity analysis that showed that NCP is robust to this parameter.
>
> Moreover, we conducted a new sensitivity analysis on the flights dataset for different noise distributions, shown in Figure 5 (see updated paper). The experimental setup is the same as for our active learning experiment. We observe that BBB+NCP is robust to the size of the input noise, which supports the conclusions we drew from the toy dataset. NCP consistently improves RMSE and NLPD of the Bayesian neural network and yields its best NLPD for all noise sizes below 0.6. For the ODC baseline, we observe a trade-off: narrower input noise increases the regularization strength, leading to better NLPD but reduced RMSE.
>
> [b) It is also interesting that noise with a shared scale is used for all 8 dimensions of the flight dataset. Is this choice mainly governed by convenience — easier to select one hyper-parameter rather than eight?]
>
> Correct, we use the same noise variance for all input dimensions. This seems sufficient because we normalize all the input dimensions to have zero mean and unit variance, as is common practice. We did not find it necessary to try other parameters for the different input channels when experimenting with NCP.
>
> [c) Presumably, the predictive uncertainties are also strongly affected by both the weighting parameter \gamma and the prior variance sigma^2_y . How sensitive are the uncertainties to these and how were these values chosen for the experiments presented in the paper?]
>
> Following a similar reasoning, we set sigma^2_y=1 since we normalize labels before training. This can be seen as an empirical prior. The scaling factor gamma for the data-space KL is generally problem dependent, analogously to the scaling factor beta for the weight-space KL. We found NCP to be a quite strong regularizer, alleviating the need for a weight-space prior (beta=0). The appendix includes an alternative derivation of NCP that sheds light on why this might be the case. We selected gamma=0.1 using a grid search over values 0, 0.01, 0.1, 1 on the low-dimensional regression task. The same parameter generalized well to the flights dataset.
>
> [d) It would be really interesting to see how well the approach extends to data with more interesting correlations. For example, for image data would using standard data-augmentation techniques (affine transformations) for generating OOD data help over adding iid noise.]
>
> This is a very interesting idea, that we are considering to explore in the future. Our main reason to focus on low-dimensional inputs in this paper is that many image tasks are classification tasks, while we are more interested in regression problems. For regression, the flights dataset is a common benchmark with published baselines.

---

### Official Review · AnonReviewer1 · 2018-11-06
**nicely written, but experiments are very limited**

**Rating:** 4
**Confidence:** 4

**Review:**

The paper considers the problem of uncertainty estimation of neural networks and proposes to use Bayesian approach with noice contrastive prior.

The paper is nicely written, but there are several issues which require discussion:
1. The authors propose to use so-called noise contrastive prior, but the actual implementation boils down to adding Gaussian noise to input points and respective outputs. This seems to be the simplest possible prior in data space (well known for example in Bayesian linear regression). That would be nice if authors can comment on the differences of proposed NCP with standard homoscedastic priors in regression.
2. The paper title mentions 'RELIABLE UNCERTAINTY ESTIMATES', but in fact the paper doesn't discuss the realibility of obtained uncertainty estimates directly. Experiments only consider active learning, which allows to assess the quality of UE only indirectly. To verify the title one needs to directly compare uncertainty estimates with errors of prediction on preferably vast selection of datasets.
3. The paper performs experiments basically on two datasets, which is not enough to obtain any reliable conclusions about the performance of the method. I recommend to consider much wider experimental evaluation, which is especially importan for active learning, which requires very accurate experimental evaluation
4. It is not clear how to choose hyperparameters (noise variances) in practice. The paper performs some sensitivity analysis with resepct to variance selection, but the study is again on one dataset.

Finally, I think that the paper targets important direction of uncertainty estimation for neural networks, but currently it is not mature in terms of results obtained.

---

> ### Author Response · Authors · 2018-11-27
> **Clarifications, differences to standard priors, new sensitivity analysis**
>
> Thank you very much for your review and the constructive suggestions.
>
> [1. The authors propose to use so-called noise contrastive prior, but the actual implementation boils down to adding Gaussian noise to input points and respective outputs.]
>
> We would like to clarify. NCP adds a term to the objective to match the epistemic variance that the model predicts for perturbed inputs to a prior value. The normal training objective using unmodified training inputs and labels is still present. While this can loosely be described as predicting noisy outputs for noisy inputs, there are technical differences. Mainly, NCP only targets the epistemic and not the aleatoric variance, thus encouraging uncertain and not necessarily noisy predictions. This encourages the model to separate epistemic and aleatoric variances, as needed to compute the information gain. Moreover, the KL term that NCP adds to the objective is computed in closed form, so there is no noise added to the outputs.
>
> [This seems to be the simplest possible prior in data space (well known for example in Bayesian linear regression).]
>
> While NCP is a natural idea, there certainly exist simpler data priors. For example, one could define a prior on the full predictive distribution, rather than targeting only the mean and epistemic variance. Moreover, one could use a data-independent prior (\mu_y=0) that might improve uncertainty but degrade generalization. We explicitly avoid more complex priors that are difficult to optimize, such as OOD GANs (Lee et al., 2017). If we missed related papers from the Bayesian linear regression literature, we would be glad to be pointed to those to discuss them. It is a desirable property that NCP is easy to implement and optimize and yet clearly improves BNNs.
>
> [That would be nice if authors can comment on the differences of proposed NCP with standard homoscedastic priors in regression.]
>
> Our reply above mentions two differences to standard homoscedastic priors in regression: NCP targets only the epistemic variance and it can be centered around training labels. In addition to defining the NCP prior, our paper contributes a practical algorithm for applying data priors to models that are optimized via variational inference. While parameter inference for linear regression has a closed form, this is not the case for BNNs. A key observation of our paper is that applying a data prior on training inputs is not enough in this case -- it must be applied beyond the training distribution if it should be enforced for unseen inputs. Our paper shows that this clearly improves the usefulness of uncertainty estimates for active learning.
>
> [2. The paper title mentions 'RELIABLE UNCERTAINTY ESTIMATES', but in fact the paper doesn't discuss the reliability of obtained uncertainty estimates directly.]
>
> While a direct evaluation of the predicted aleatoric and epistemic variances would be ideal, doing so quantitatively is difficult for models without closed-form posterior and multi-dimensional datasets. We decided for active learning experiments to go beyond the mainly qualitative analysis often conducted for new OOD methods. Active learning using the expected information gain is specifically appropriate in our case, as the noise and uncertainty predictions take opposing roles in this acquisition function. We point to Figure 1 for a qualitative evaluation on the 1D toy dataset that can be visualized. If the comment is mainly directed at the paper title, we are open to changing it to emphasize active learning.
>
> [3. The paper performs experiments basically on two datasets, which is not enough to obtain any reliable conclusions about the performance of the method. I recommend to consider much wider experimental evaluation, which is especially importan for active learning, which requires very accurate experimental evaluation]
>
> We agree a wider experimental evaluation would further strengthen our conclusions. While more tasks are always desirable, we would like to point out that prior methods to improve uncertainty estimates have often been demonstrated on two datasets. For example,  Gal et al. (ICML 2017) train on the MNIST and ISIC datasets and Lee et al. (ICLR 2018) train on the SVHN and CIFAR-10 datasets.
>
> [4. It is not clear how to choose hyperparameters (noise variances) in practice. The paper performs some sensitivity analysis with resepct to variance selection, but the study is again on one dataset.]
>
> We conducted a new robustness analysis on the flights dataset for different noise distributions, shown in Figure 5 (see updated paper). We observe again that BBB+NCP is robust to the size of the input noise, supporting the conclusions from the toy dataset. NCP consistently improves RMSE of the BNN and yields its best NLPD for all noise sizes below 0.6.
>
> We hope these comments clarified our paper submission and its connections to typical priors in regression.

---

### Official Review · AnonReviewer2 · 2018-11-07
**An interesting approach to quantify uncertainty in neural networks**

**Rating:** 7
**Confidence:** 3

**Review:**

This paper presents an approach to obtain uncertainty estimates for neural network predictions that has good performance when quantifying predictive uncertainty at points that are outside of the training distribution. The authors show how this is particularly useful in an active learning setting where new data points can be selected based on metrics that rely on accurate uncertainty estimates.

Interestingly, the method works by perturbing all data inputs instead of only the ones at the boundary of the training distribution. Also, there is no need to sample outside of the input distribution in order to have accurate uncertainty estimates in that area.

The paper is clear and very well written with a good balance between the use of formulas and insights in the text.

The experimental section starts with a toy 1d active learning task that shows the advantage of good uncertainty estimates when selecting new data points. The authors also present a larger regression task (8 input dimensions and 700k data points in the training set) in which they obtain good performance compared to other models able to quantify epistemic uncertainty. In my opinion, the experiments do a good job at showing the capabilities of the algorithm. If anything, since the authors use the word "deep" in the title of the paper I would have expected some experiments on deep networks and a very large dataset.

---

> ### Author Response · Authors · 2018-11-27
> **Accurate summary, removed word deep from title**
>
> Thank you very much for your review.
>
> [Interestingly, the method works by perturbing all data inputs instead of only the ones at the boundary of the training distribution.]
>
> Correct, this is likely caused because the standard training objective outweighs the NCP loss inside of the training distribution. It is analogous to implicit priors for neural networks such as weight decay, which are also outweighed by the prediction loss inside of the training distribution.
>
> [Also, there is no need to sample outside of the input distribution in order to have accurate uncertainty estimates in that area.]
>
> Adding noise to the inputs often results in inputs that are outside of the training distribution. As you pointed out, our experiments indicate that it is enough to apply the prior on inputs near the training distribution.
>
> [The experimental section starts with a toy 1d active learning task that shows the advantage of good uncertainty estimates when selecting new data points. The authors also present a larger regression task (8 input dimensions and 700k data points in the training set) in which they obtain good performance compared to other models able to quantify epistemic uncertainty. In my opinion, the experiments do a good job at showing the capabilities of the algorithm.]
>
> Thank you.
>
> [If anything, since the authors use the word "deep" in the title of the paper I would have expected some experiments on deep networks and a very large dataset.]
>
> Thank you for this suggestion. We agree that removing the word "deep" makes the paper title more descriptive.

---

### Public Comment · ~Andrey_Malinin1 · 2018-10-10
**Related work**

Hello! :) Interesting work. You may find our work on predictive uncertainty estimation to be relevant.

https://arxiv.org/pdf/1802.10501.pdf

---

> ### Author Response · Authors · 2018-11-27
> **Interesting paper!**
>
> Thank you for pointing out your related recent work.

---

### Meta-Review · Area_Chair1 · 2018-12-12
**Limited experiments**

**Confidence:** 4
**Recommendation:** Reject

**Metareview:**

The paper studies the problem of uncertainty estimation of neural networks and proposes to use Bayesian approach with noice contrastive prior.

The reviewers and AC note the potential weaknesses of experimental results: (1) lack of sufficient datasets with moderate-to-high dimensional inputs, (2) arguable choices of hyperparameters and (3) lack of direct evaluations, e.g., measuring network calibration is better than active learning.

The paper is well written and potentially interesting. However, AC decided that the paper might not be ready to publish in the current form due to the weakness.